# (In)Visible illness: A photovoice study of the lived experience of self-managing rheumatoid arthritis

Susie Donnelly[1,2]*, Anthony G. Wilson[2], Hasheem Mannan[1,3], Claire Dix[4], Laura Whitehill[2], Thilo Kroll[1]

**1** Centre for Interdisciplinary Research Education and Innovation in Health Systems (IRIS), School of Nursing, Midwifery and Health Systems, University College Dublin, Dublin, Ireland, **2** School of Medicine and Medical Science, Conway Institute, University College Dublin, Dublin, Ireland, **3** FLAME University, Pune, India, **4** Independent Filmmaker, Dublin, Ireland

\* susie.donnelly@ucd.ie

## Abstract

### Background

Chronic illnesses, such as Rheumatoid Arthritis (RA), are a growing burden on health care systems worldwide. Self-management emphasises the patient's central role in managing their illness. This is pertinent given the majority of care is provided by the individual themselves; yet how individuals make sense of self-management in everyday life is largely unseen.

### Objective

The purpose of this study was to capture the strengths and concerns of individuals with RA in self-managing their illness, raise awareness of their lived experience and spark a dialogue among stakeholders.

### Methods

A community-based participatory approach, Photovoice, was adopted. A purposive sample of participants were tasked with taking photographs to represent the challenges and solutions to living with RA. Group workshops and semi-structured interviews were conducted to facilitate reflection, dialogue and analysis. Data analysis followed Braun and Clarke's thematic analysis. Public exhibitions were held throughout the Autumn of 2019.

### Results

Eight women and three men (n = 11) across suburban and urban regions of Ireland were recruited (mean age 57 years, disease duration 4–21 years). Participants identified four main themes which reflected the lived experience of self-managing RA: (i) I'm Here but I'm Not, (ii) Visible Illness, (iii) Medicine in All its Forms, (iv) Mind Yourself. These themes captured the challenge of reduced agency, limited contribution and participation, and a complex

---

**Data Availability Statement:** All data underlying the findings of the manuscript (e.g. interview transcripts is available from the Irish Qualitative Data Archive (IQDA) / Digital Repository of Ireland

(DRI) for teaching and research purposes (https://doi.org/10.7486/DRI.8c980c55x).

**Funding:** Funding for this research was awarded to SD under a Wellcome Trust Institutional Strategic Support Fund (ISSF) which was financed jointly by University College Dublin and the SFI-HRB-Wellcome Trust Biomedical Research Partnership as part of a Medical Humanities and Social Science Collaboration Scheme (Grant number 204844/Z/16/Z). This fund provided support in the form of salaries for authors CD and SD, but did not have any additional role in the study design, data collection and analysis, decision to publish, or preparation of the manuscript. The specific roles of these authors are articulated in the 'author contributions' section.

**Competing interests:** The authors have read the journal's policy and declare the following competing interests: A Wellcome Trust Institutional Strategic Support Fund (ISSF) financed jointly by University College Dublin and the SFI-HRB-Wellcome Trust Biomedical Research Partnership provided support in the form of salaries for authors CD and SD. There are no patents, products in development or marketed products associated with this research to declare. This does not alter our adherence to PLOS ONE policies on sharing data and materials.

relationship between visible and invisible illness. Solutions focused on improving psychological and emotional resilience, particularly through personal reflection and increased agency.

## Conclusions

Our findings suggest that RA is experienced as a fluid relationship between states of masking and surfacing of illness shaped by contextual and situational factors. Photovoice was a highly effective tool to capture and communicate this complexity. Supporting increased agency among individuals with RA to control the (in)visibility of illness and disability can inform the development of future self-management support.

## Introduction

Chronic illnesses—such as cancer, diabetes, respiratory disease and arthritis—are a growing burden for healthcare systems worldwide and are a prevalent cause of mortality and morbidity [1–3]. Rheumatoid arthritis (RA) is a chronic autoimmune disease estimated to affect between 0.5–1.1% of the Northern European and Northern American populations [4]. It is characterised by inflammation of the joints causing pain, stiffness, and swelling. Without effective treatment, it is associated with deformity of the joints (typically hands and feet) leading to disability, decreased quality of life, and increased mortality [5–9]. For most, it is a life-long disabling condition with no cure.

RA can be characterised as an illness that generates both visible (e.g. deformity and swelling) and invisible (e.g. fatigue, pain) disabilities. The United Nations Convention on the Rights of Persons with Disabilities [10] recognises disability as 'an evolving concept'. It asserts that disability manifests from the interaction between a person with impairment and societal barriers (such as negative attitudes or obstructive environments) which results in unequal participation. Zhao et al. [11] illustrated that pain was positively associated with functional disability in RA patients. The experience of depression, social isolation, as well as the cognitive and emotional adjustments to RA are similarly invisible [12, 13]. In studying Korean patients with RA, Uhm et al. [14] reported that functional disability and depression, rather than disease activity and pain, have profound effects on health-related quality of life. Chiou and Huang [15] reported that older patients with rheumatoid arthritis had significantly higher levels of disability, disease activity, more depression and less life satisfaction than patients with osteoarthritis. Importantly they indicate subjective and available social support as the protective predictors for functional disability.

Invisible symptoms are generally subjective making it hard for others to understand and empathise, and can be difficult to clinically measure [16, 17]. This can undermine the validity and legitimacy of the person's illness as distinctly medical; a concept understood as "contested illness". This can lead to a questioning of the authenticity of the person's symptoms and their mental health [18]. Thus, people with RA can struggle with feelings of not being believed, particularly from family, colleagues and social services which in turn may impact their mental well-being and social functioning [19]. Conrad and Barker contend that invisibility 'becomes one of the most important characteristics of these illnesses, affecting access to a diagnosis and health care, the response of others to one's problem, and the very identity of the sufferer' [18, p.S70]. Since the 1990s, the Chronic Disease Self-Management Program (CDSMP) developed by the Stanford Patient Education Research Center has become an increasingly popular intervention for patients and aims to enable people with chronic diseases to better manage their

illness and achieve greater self-efficacy [20–23]. The CDSMP and other self-management programs can be community or hospital based and focus on improving patient's health behaviour through skill mastery, improved confidence and goal setting. While it was initially developed with RA patient cohorts, it has become a standardized program applied across populations and societies [24, 25]. Although it has demonstrated benefits, the fidelity of the intervention across cultures, programme retention and recruitment, and its effectiveness in improving patient outcomes in the long-term requires further study [25–29]. Arguably, the lived experience of self-managing chronic illness and the context in which effective self-management behaviours are adopted—or not—requires further exploration.

Since the 1980s, health and social researchers have increasingly using qualitative methods to explore the subjective, experiential aspects of chronic illness in order to improve care, understanding, and public policy [30, 31]. Thus far, qualitative studies have generated rich, deep knowledge of the lived experience of RA and offer valuable frameworks and themes to demonstrate how people living with RA maintain a functional role in society [32–36]. However, it is still unclear how themes such as disease symptoms, control of one's body, and emotional challenges, are applied in everyday life. Narrative methods have provided good insights into the perceptions, beliefs, attitudes and reported behaviours of individuals living with arthritis and other long-term conditions [36]. However, data and information obtained through interviews alone may miss some of the nuances of lived experience that either may be less salient in a person's mind or cannot easily be expressed by verbal means alone [37–40]. Moreover, communication and personal expression has become increasingly visual over the past two decades with the introduction of digital photography and social media for the sharing of visual information. In social research, the collection, analysis and dissemination of data increasingly draws on visual methodologies [41, 42].

Photovoice is a highly participatory, action research approach whereby people document their everyday realities through photography. It is rooted in values of human creativity and democratic participation and informed by critical pedagogy, feminist theory and photojournalism [43–46]. Photovoice was developed in the 1990s by Caroline Wang and Mary Ann Burris as a health promotion tool for rural women in China whose living and working conditions were, until then, not well understood by local policymakers [46]. By providing people in the community with cameras, photovoice enabled them to '1) record and reflect their community's assets and concerns, 2) discuss issues of importance to the community in large and small groups to promote critical dialogue and produce shared knowledge, and (3) reach policy makers' [47, p. 560]. It is especially well-suited to the study of invisible illness as a means of exposing the unseen everyday realities and hidden conditions of people's experiences [48, 49].

Photovoice is an established methodological tool in health research to explore marginalised experiences, such as disability, impaired mental health as well as chronic fatigue and pain [50–53]. The methodology has been reported to contribute to enhanced understanding of the needs of communities, and greater empowerment of these communities [54–56]. While the method is user-friendly and can generate particularly rich data, the rigour with which it is applied is open to inconsistency and thus its effectiveness may vary [37, 57]. The method can also present challenges for participants in terms of understand expectations around the photography task and navigating issues such as the consent of photographic subjects [58]). At the individual patient level, there has been an acknowledgement that patients should be involved in decisions about their treatment to ensure better adherence to treatment resulting in improved outcome [59]. However, how patients can be assisted to be meaningfully involved in the decision-making process is less straightforward [60]. The aim of this paper is to understand how individuals (self-)manage life with rheumatoid arthritis through their involvement in visual and narrative means.

## Method

### Study design

This study followed the procedure for conducting photovoice studies outlined by Wang and Burris [46]. The collected qualitative data comprised of semi-structured interviews, photographs, ethnographic field notes and observational notes and discussion protocols from two group workshops. Basic sociodemographic information and related data were collected during the screening and recruitment phase (Table 1).

This study was conducted by a multi-disciplinary academic team in collaboration with a Research Advisory Group (RAG) and study participants. [**SD**] is a postdoctoral fellow in health systems research with a PhD in sociology and a background in journalism. She has used photovoice in prior chronic illness research. The author [**AGW**] has extensive experience in the genetics and clinical practice of rheumatology. The author [**HM**] is a global health researcher with subject knowledge in disability policy. The author [**CD**] is a filmmaker and visual artist concerned with marginalised groups and subcultures and has collaborated with social scientists using participatory methods. The author [**LW**] has a background in sociology and training in general medicine. The final author [**TK**] is a health systems researcher with extensive experience in applied qualitative research and personal and subject knowledge in relation to arthritis and public and patient involvement.

### Inclusion criteria and recruitment

A purposive recruitment procedure was carried out using maximum variation sampling to obtain a heterogeneous study group, and thereby a rich variation in results. Individuals were chosen with respect to age, gender and employment status. We wanted to explore self-management beyond the established educational models (such as the Chronic Disease self-management Programme), therefore anyone who had attended a formal self-management education programme was excluded. This also meant that our sample reflected a population who were not primed by formal self-management discourse and could provide original insight into the lived experience of self-managing RA.

Over approximately three months, two recruitment strategies–clinical and public—were used simultaneously to obtain a sample of eleven participants. These two strategies were used to improve response rates and the heterogeneity of the sample. Four participants were recruited from a hospital setting where they were approached by a dedicated rheumatology Health Care Professional (HCP) with a follow-up call from the researcher [**SD**] to confirm

**Table 1. Sample demographics and characteristics.**

|               | Sex    | Age | Paid Employment | Years Since Diagnosis | Comorbidities |
|---------------|--------|-----|-----------------|-----------------------|---------------|
| Participant A | Female | 37  | Yes             | 4                     | Yes           |
| Participant B | Female | 38  | Yes             | 4                     | Yes           |
| Participant C | Female | 41  | Yes             | 5                     | No            |
| Participant D | Female | 43  | Yes             | 4                     | No            |
| Participant E | Male   | 49  | No              | 4                     | Yes           |
| Participant F | Female | 61  | No              | 21                    | Yes           |
| Participant G | Female | 66  | No              | 11                    | Yes           |
| Participant H | Female | 70  | Yes             | 9                     | No            |
| Participant I | Male   | 70  | No              | 5                     | No            |
| Participant J | Female | 77  | No              | 7                     | No            |
| Participant K | Male   | 83  | No              | 20                    | Yes           |

eligibility. Seven participants responded to public calls for study volunteers and actively approached the researcher by email or phone. Advertisements were placed in regional newspapers, community newsletters, and social media as well as leaflet drops in a number of local cafes and community noticeboards. Additionally, community-level patient groups were approached with the support of the national arthritis advocacy organisation, *Arthritis Ireland*. Of the seven participants who responded to public advertisements, one responded to a regional newspaper advert, one came across a flyer, and five responded to online recruitment calls such as e-newsletters from associated patient groups or social media platforms. Two sample images were included in the Participant Information Leaflet (PIL) (**S1 Appendix**). These images were taken by the lead researcher and a member of the RAG. They were intended to be informative about the intention of the photo assignment while avoiding priming participants. The language and content of the PIL and consent form was reviewed and agreed with the RAG and followed guidance set out by the relevant university and hospital ethical boards.

## Procedure

Fieldwork involved group workshops, individual semi-structured interviews and public exhibitions, as shown in Fig 1. Fieldnotes regarding participants' group interactions and various aspects of the research process were maintained [**SD**]. The research team met monthly to discuss emerging themes and review fieldwork, and participants were involved in informing and validating findings at various stages.

To support an effective small-group discussion the sample was randomly split into two groups for the delivery of the workshops; Group A (n = 7), Group B (n = 4). Each workshop was three hours in duration and delivered jointly by a researcher [**SD**] and visual artist [**CD**]. A 3-hour session was deemed adequate to cover the objectives of each workshop (**S2 and S3 Appendices**).

**Group workshop 1.** The first workshop focused on introductions, an overview of the project, exercises in visual literacy [61] and how to use the cameras. Participants were given a photo assignment titled, "The challenges and solutions to living with RA" and potential ethical issues were discussed, such as taking images of identifiable people and places, as well as copyright and ownership of the photographs (**S2 Appendix**).

**Interviews.** Approximately two weeks later, individual semi-structured interviews were conducted with participants at their home or a similarly private and non-clinical location

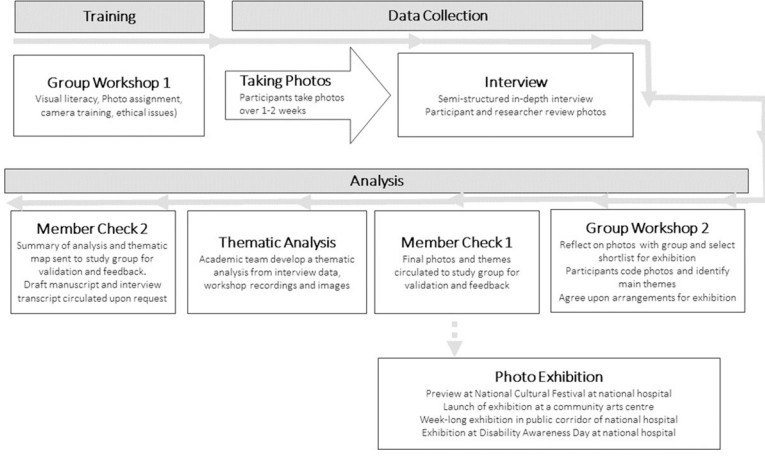

**Fig 1. Phases of data collection and analysis.**

convenient for the interviewee. When scheduling these interviews, the lead researcher (SD) used the opportunity to check how the participants were progressing with the phot assignment and answer any questions they had. During the interview, participants were asked to share approximately five photographs that best represented the challenges and solutions to living with RA. In reviewing the photos, the SHOWED technique was used to elicit analysis and reflect upon possible action, as follows: "What do you **S**ee in this picture? What is actually **H**appening in the picture? How does this relate to **O**ur lives? Why does this situation, concern or strength **E**xist? What can we **D**o about it?" [62: 188]. Interviews were conducted by [**SD**] and where requested, creative support for the photo assignment was provided remotely by [**CD**]. Interviews were transcribed by [**LW**]. The average duration of interviews was 1 hour and 13 minutes (ranging from 54 minutes to 1 hour and 46 minutes). Participants were invited to review their verbatim interview transcript.

**Group workshop 2.** At the second group workshops, participants were given printed copies of the photos they had discussed during the interviews, as well as any additional photos they had sent to the researcher for inclusion in the interim (**S3 Appendix**). Photographs were disturbed in sealed envelopes and participants were advised to only to share the images they wished. They were asked to share and discuss their photos in small groups of two. Each participant then presented a selection of their photos to the wider group for general discussion. A mini-exhibition was assembled on the walls of the room and participants were asked to vote, using stickers, for the photographs they wished to be included in a public exhibition. The shortlisted photographs were then clustered into similar groupings and participants began to describe potential themes represented by these clusters. The workshop with Group A was run first and therefore they were the first group to identify potential themes. In the workshop with Group B, the same procedure was followed and they produced a set of themes. They were then asked to consider their themes and images in tandem with those of Group A. Collectively, the participants in Group B, the researcher (SD) and visual artist (CD) discussed how to integrate the themes and images of both groups. As a form of "member checking" to improve the trustworthiness of these themes and to validate results [63], shortlisted photos, captions and themes were circulated to all participants by email or post for feedback and agreement before the exhibition.

Details of the exhibition, such as the day, location and invitation list were discussed with workshop participants in both groups. It was agreed that a sub-committee composed of one representative from each group would be formed to assist [**SD**] and [**CD**] with finalising and curating the photo exhibition.

**Exhibition.** Throughout the Autumn of 2019, a series of exhibitions of the photographs was held. A day-long public photo exhibition was launched at a community arts centre in Dublin, followed by a week-long exhibition at a general public hospital, also in Dublin. A digital preview of the exhibition was held as part of a national cultural night. Finally, a mini-exhibition was also featured as part of disability awareness initiative at the hospital. These events were attended by policymakers, representatives from community and voluntary organisations, clinicians and health care professionals, researchers across social and biomedical sciences, and members of the general public. It was also attended by members of the study group and research advisory group, often with family members and friends.

## Data analysis

In the second group workshops, participant acted as co-researchers to inductively develop four analytical themes representing the challenges and solutions to living with rheumatoid arthritis. These were then deductively applied by the academic researchers to the rest of the

data (e.g. interviews and photos). These workshops were audio recorded and transcribed for veracity. In analysing the interviews, Braun and Clarke's [64] phases of thematic analysis (**S4 Appendix**) were followed. Subthemes were identified within the four main themes generated by participants. [**SD**] performed a "live coding" which involved manual coding while simultaneously listening to the audio recording of each interview. This was done to further support the researcher's understanding of the intent, context, and meaning of the words and images [65]. A combination of traditional material methods (e.g. coloured pens, paper, and sticky notes) were combined with the use of software (NVivo 12) for additional coding, data organisation and mapping. Maher et al. [66] advocate this grounded practice to maximise the researcher's interaction with the data and ensure the analysis process is rigorous and productive. The researcher drew from interaction design thinking by adopting affinity diagrams to identify and cluster themes [67, 68]. Members of the research team [**SD**], [**TK**], [**LW**], [**HM**] met to review and hone the development of subthemes and thematic mapping. A summary of the analysis and thematic map was circulated to participants by email who were invited to provide clarification and commentary via email or freepost [69]. A face-to-face workshop to validate and member-check results with the Research Advisory Group and study participants was cancelled due to the onset of the Covid-19 pandemic. The Consolidated Criteria for Reporting Qualitative Research (COREQ) checklist is reported in **S5 Appendix** [70].

## Ethical considerations and data protection

This study was approved by institutional research boards at the Human Research Ethics Committee–Sciences, University College Dublin (LS-18-66-Donnelly), and the Institutional Review Board, Mater Misericordiae University Hospital, Dublin ((Ref: 1/378/2017). Besides obtaining written informed consent from all participants, ethical issues—particularly around how the photographs are used—were considered and reflected upon at various stages to ensure confidentiality, respect, and equity of participation [71]. Drawing from prior reflections of the impact of photovoice and participatory action research [72, 73], we endeavoured to manage participant's expectations around the impact of our research on policy change, the time taken to publish findings, and the likelihood of future research [73]. Written release forms were obtained for all photos used for publications. For long-term preservation of the data, de-identified data (i.e. interview transcripts, captioned images) were archived permanently with a national qualitative data archive for use in future teaching and research [74]. Following best practice guidance on payment and recognition for patient and public involvement [75], attendees at any study related meetings were gifted a €25 ($28) voucher to contribute towards the cost of travel and related expenses. Depending on the duration of the meeting, a light lunch or refreshments were provided. Basic digital cameras (Polaroid IX828 20MP Zoom Compact Camera; CamKing CDC3 2.7" TFT LCD HD Mini Digital Camera) were also gifted to each participant to use in the project and keep afterwards.

**Public and patient involvement.** At each stage of the research process, participants made decisions about the aspects of their experience they wished to share. They decided how, when and with whom their data (e.g. interview transcripts and photos) were shared. In line with best practice frameworks for Patient and Public Involvement (PPI), this study used consultation and development throughout the design [76]. This was done to ensure the process was open, transparent and democratic and that people with lived experience were actively involved in the research process at all levels. The academic research team included two authors with lived experience of self-managing chronic illness, including arthritis [**TK**] and autoimmune disease [**SD**].

In addition, a Research Advisory Group (RAG) was established at the outset of the study to inform its design and execution. This group was composed of five volunteers with lived

experience of RA. A series of meetings were held where various aspects of the project were presented and feedback was offered. Members of the RAG shared journey maps of their illness, provided written feedback on drafts of flyers and participant information leaflets, and informed the phrasing of the photo assignment. They also assessed the camera equipment used in the study for accessibility and ease of use.

# Results

## Study participants

We recruited a total of eleven adults, 18 years and older with a clinical diagnosis of rheumatoid arthritis (eight women and three men). The participants' ages ranged from 37–83 years at the time of recruitment (average 57.73 years) living in urban or suburban areas [77]. Five participants held paid employment, including one participant on maternity leave and another on a career break. Six participants had comorbid chronic conditions, these included other musculoskeletal conditions, respiratory diseases, hypothyroidism, tinnitus, hypertension. Participants had been living with an RA diagnosis between 4–21 years at the time of the recruitment (average 8.55 years). Five participants were being treated by a private consultant, while the remainder received treatment within the public healthcare system.

## Visual and narrative themes

Participants produced 122 photographic images in total (ranging from 1 to 22, average 11). The photos were discussed and reflected upon during individual interviews and in group workshops. Following this process, a series of four themes were generated by participants: *I'm Here but I'm Not*, *Visible Illness*, *Medicine in All its Forms*, and *Mind Yourself* (See Fig 2). Participants selected a total of 32 photos to represent these themes in a series of public exhibitions. [**CD**] and [**SD**] ensured that each participant had at least one photo included. It is unlikely that through further fieldwork or an increased sample size, new themes would have emerged that had not already been discussed, therefore data saturation was reached [78].

## I'm Here but I'm Not

When describing the theme, *I'm Here But I'm Not*, participants spoke about a sense of living in the background as a result of their illness and a feeling that life was going on around them but they were not part of it. The images in this category often represented a potential life unlived (See Figs 3 and 4).

Fig 4 shows a participant's husband and two children playing at the seashore. On this particular morning, the participant woke up with severe pain which she explained prevented her from holding the hand of her three-year-old child. She sat on the beach and took this photograph while watching her family. She expressed how, upon reflection, this scene provoked in her a strong sense of anger towards the disease and the limitations it imposes. Her experience and limitations were invisible, as she explains '*anyone walking past would have seen a Dad having a great time with his children*'.

**You're not doing enough.** Participants expressed a desire to contribute to work, society and family life. They wished to contribute within their various roles as partners, parents, employees, friends and so on but often struggled with a sense of failure or falling short: '*I'm not helping enough, I'm not doing enough*'. They wanted to be productive, and to be valued by others. For example, when asking someone for help with tasks, some participants explained a reciprocal relationship which reduced their sense of burden; '*I'm very lucky that [my family] are as ready and willing to support and help as much as they are. But I guess, I've done it for*

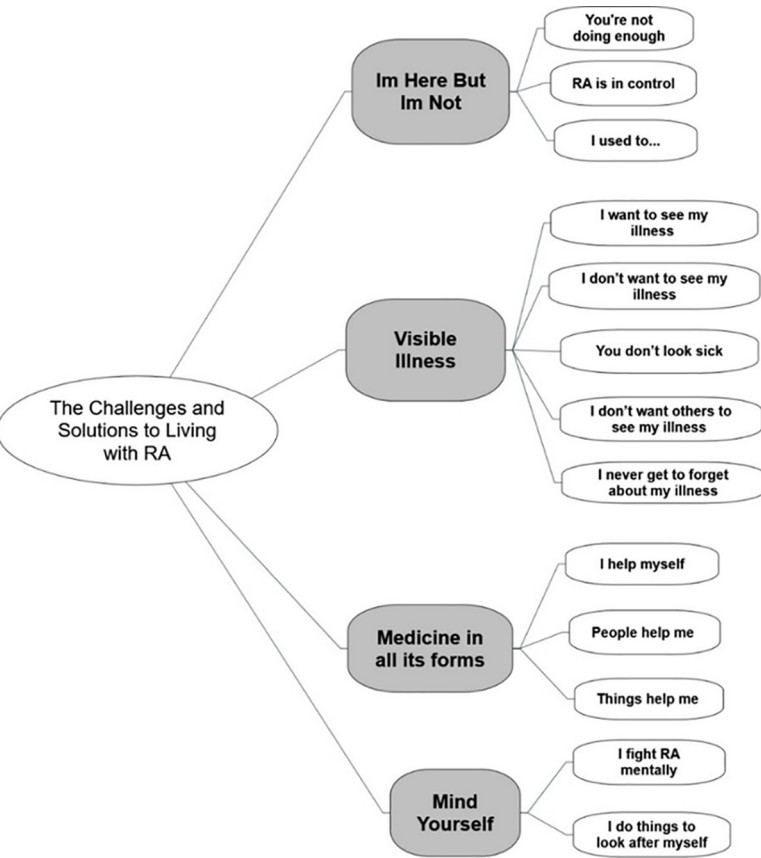

**Fig 2. Thematic map.**

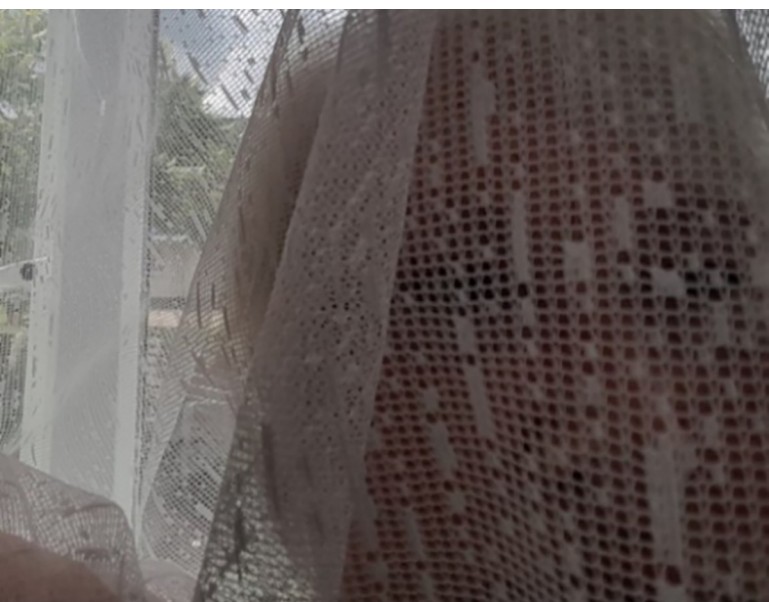

**Fig 3. Too tired.** Can't connect. This is a photograph, a bad photograph, I'd like to do a better photograph of me being behind a veil. Or a screen trying to communicate how I am. And people not coming through to other people. I wanted to do something to depict that. Will they listen? Will they hear?

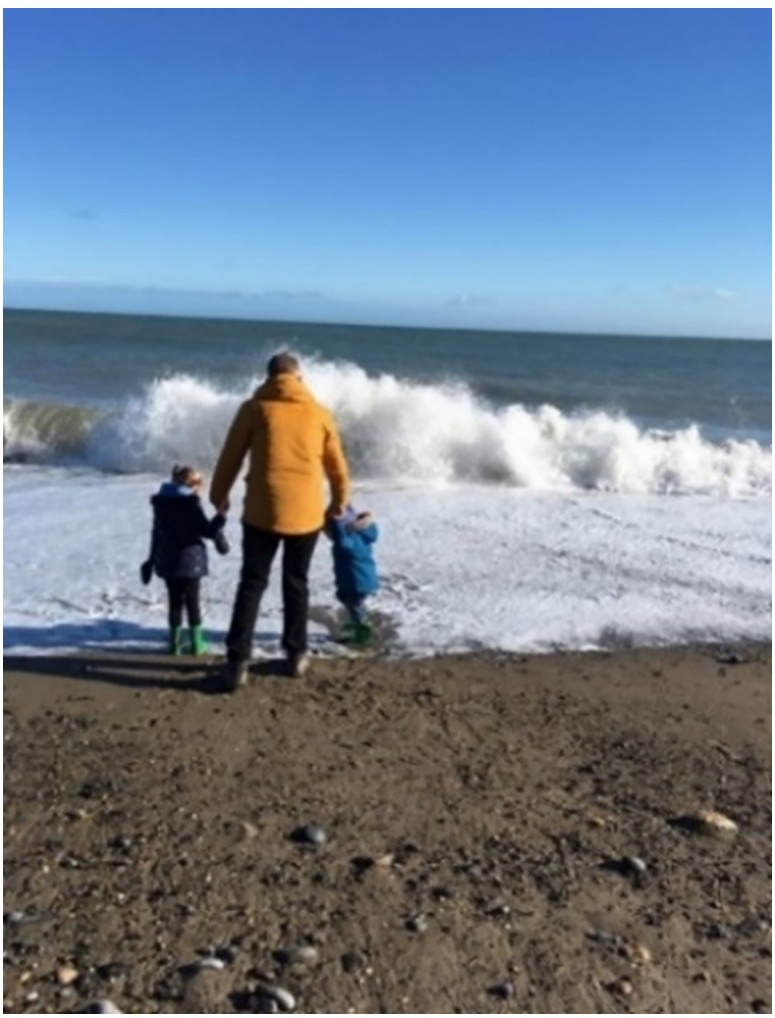

**Fig 4. Lost at sea.** With RA every morning when you wake up, the first thing I find I do is see what number I'm at on the pain scale and then is it a "hard" 8 or an easy "8"- always influenced by whether or not you've managed to get a full night's sleep on the pain front. On this particular day it was hard 8, one of those days where you really don't want to do anything other than simply cope with the pain and the unrelenting unremitting nature of it as best you can. BUT you're a person who is more than just their disease, you're a wife and mother. You have obligations. You get up, get dressed and head to the sea side for the fun day out you promised your kids. On a day that's a hard 8 though, your hands, fingers, wrists, are all too sore to be able to manage holding hands with your children. Imagine, holding hands with your 3 year old hurts too much? It's a hard 8 all right. So I sat with my baby in her buggy and watched my children have such fun with their dad as they ran, jumped, splashed, got soaked and enjoyed their lives but wondered why I wouldn't or couldn't join in. The photo shows a sea of two halves for me. A beautiful calm sea far out, sun shining, and anyone walking past would have seen a dad having a great time with his children. For me I see the bit where the waves are crashing and breaking against the shore- the force of the waves represent my anger and rage at the disease, the havoc it has brought into my life. The crashing and breaking of the waves is often how I feel the disease attacks my body- it's full on and pulls no punches. At times RA pulls you under like it's a rip tide and before you know it you're lost at sea.

*them'*. A number of grandparents in the study spoke of the frustration and guilt of not being able to hold their grandchildren on their lap. Mothers reported not being able to cradle their babies. Most participants expressed feelings of guilt around uneven distribution of domestic labour. The ability to contribute to a relationship as an equal partner, as well as fears of becoming a burden, were a concern:

Are they [potential partner] going to feel like I'm holding them back? Are they going to feel like I'm a burden? I mean it's ok now, but what if I get worse and they have [to help]. I saw what my Dad had to give up to take care of my mom, so I just think, am I going to do that to someone? You know in 10, 15, 20 years from now? . . .I know medicine is advancing and it's getting better and the chances of that are slimmer, but you think about those things. And you think [of]. . . the financial responsibility. What if I lost my job and I have all these medical things I have to have?

In the context of employment, being perceived by colleagues or managers as "lazy" was a concern. A participant articulated her fears around how she might be regarded by colleagues if she were to take sick leave: *"I'm going to be missing from work and people are going to say* "Where is she? *She must be sick again? [They would be thinking] I was a bit of slacker"*. Being seen as unproductive or less able to contribute could be perceived as weakness, as another participant believed his illness could be exploited:

[You] try to cover up the pain, and not let people see that you were in pain. Because if you were in pain, you were susceptible for things to happen, like people taking work from you, like taking advantage of your disadvantage.

The lack of contribution to society generated feelings of isolation and alienation among some participants. They described the idea of being "lonely in a crowd" to capture these feelings, as illustrated in Fig 5 depicting an orange placed in an apple tree.

**RA is in control.**   Participants expressed diminished control (agency) over their lives as a result of their illness. Although they had choices in their everyday lives, they reported their choices were often limited as a result of their illness. As one participant describes, these limitations need to be evaluated in the most everyday decision-making:

I'm trying to get healthy and then the other side of me is saying, look where you're at? Would you not just go out and buy yourself some biscuits? And I know I do it, but I know my rheumatoid arthritis isn't going to thank me for it. My body is not going to say Ah! I just can't do [it]. I do whatever everybody else is doing but I haven't got the same freedom to do it.

**I used to**. . .. Participants spoke about the activities they used to take part in, such as sports or travel. Comparisons were made between what they could do now, and what they did in their "old life", prior to the onset of symptoms. Participation in many activities from daily tasks to social and cultural events were impacted by illness. For example, footwear was a source of frustration and a reminder of illness for many female participants. They expressed feelings of not "fitting in" at social events. If they chose to adhere to cultural norms by wearing high-heeled shoes, pain, swelling and immobility were often the trade-off (Figs 6 and 8).

In Fig 6, high-heeled shoes are marked with sticky notes: Pre-RA Work, Pre-RA Wedding, Pre-RA Night Out, Pre-RA Funeral. Trainers are marked as: Post-RA Work, Post-RA Wedding, Pre-RA Night Out, Pre-RA Funeral. The photographer is capturing the progression of the disease over time, rooting it in specific cultural and social contexts. Her self-identify is redefined from an "old life" prior to the onset of RA, to a new life as a person who is coming to accept her illness.

## Visible illness

The participants initially identified the theme, visible illness, to capture a range of images which they believed evidenced their illness.

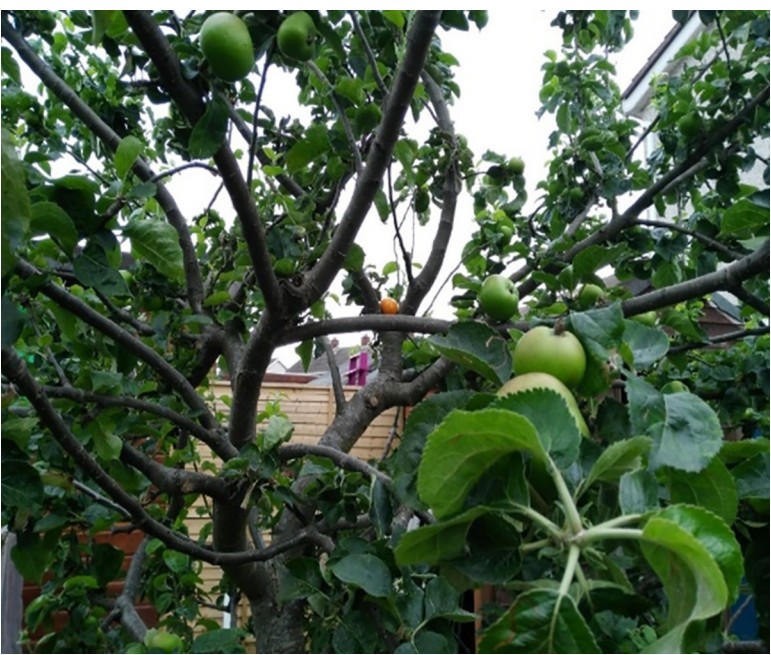

**Fig 5. Orange in apple tree.** The orange out of place in the trees surrounded by apples. It's how you start to. . .feel and question yourself in society sometimes. Just technically [you're] the same but you're very, very different. I think you feel sometimes, when you get the rheumatoid arthritis, how then [do you] fit into society? It seems for me [its] not working anymore. Even simple choices. I saw somebody [on the bus] being. . .really drunk and obnoxious [to two girls]. I wanted to get up and get them off. But it's a realization I couldn't do anything about it. . .with breathing, and the weakness, and the RA. And suddenly you start questioning yourself and looking at that picture, it makes me think when you're out of place and then trying to redefine your place in society.

**I want to see my illness.** Many participants took photographs–typically of their hands and feet—to illustrate their disease activity; such as deforimity, shown in Fig 7 or swelling. during a flare, as shown in Fig 8. These were visible markers of the progression of the illness.

In Fig 8, the photographer explains her feet became swollen as a result of wearing high-heeled shoes at a social outing. The swelling of her feet was a curiosity. These kinds of photographs enabled the person with RA to measure and understand disease activity in a way that was meaningful for them. Routine activities, particularly in the morning when stiffness is increased, could signal the presence of the illness (Fig 9).

The ability to measure the progression of the illness was important. As well as observing swelling, participants would observe the time it took to complete a routine walk in the park or to church to see if their mobility was improving.

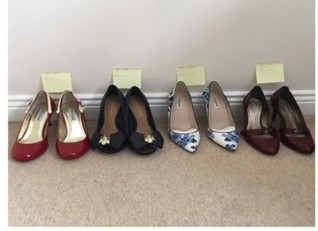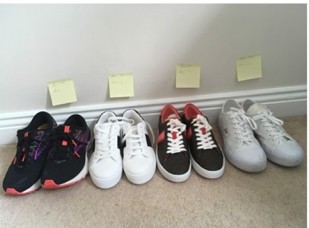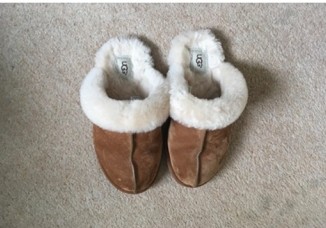

**Fig 6. Steps of acceptance.** A loss of self and a loss of identity describe what I see when I look at these shoes.

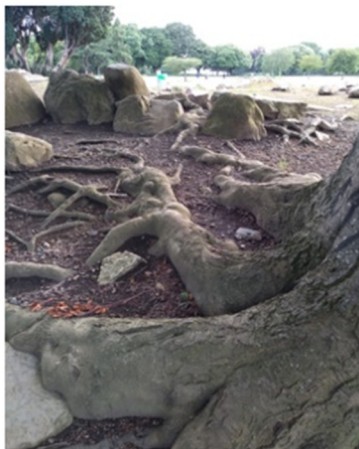
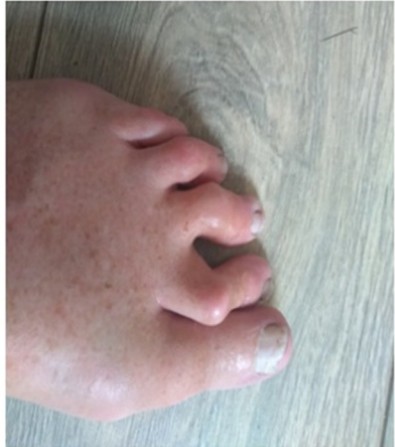

**Fig 7. The multiple…with knobs on.**

**I don't want to see my illness.** While observing indicators of disease activity and progression could be beneficial, this practice could also be distressing and demoralising. These feelings were typically provoked by deformity in the hands and feet, described as "ugly", and represented increased disease activity as well as serving as a reminder of the presence of the illness and a "sick" identity (Fig 10).

**You don't look sick.** The concept of contested illness was expressed. As one participant explains: '*You might be feeling [bad] but people say "Oh you look great". You're sore. You're in pain. Your back is at you. Your hip is at you, [but] you look great, so you must be*'. Many participants struggled to achieve recognition of their symptoms. While photographs of swelling served to validate symptoms to the patient themselves, they could also be used as evidence to others, particularly Health Care Professionals (HCPs) (Fig 11).

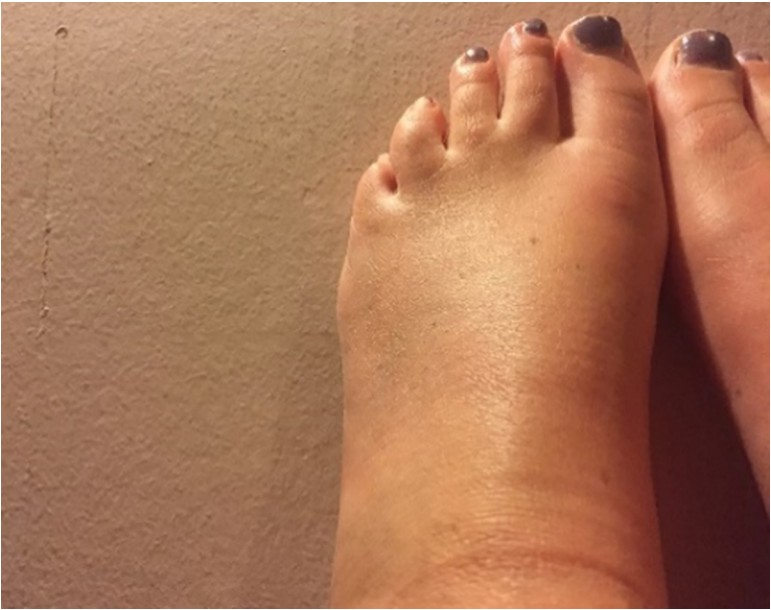

**Fig 8. Curious swelling.** Well I always find things interesting, so I'm always like, would you look at that? Because…that's not painful. That's more just curious, kind of seeing where [the swelling] goes up to…

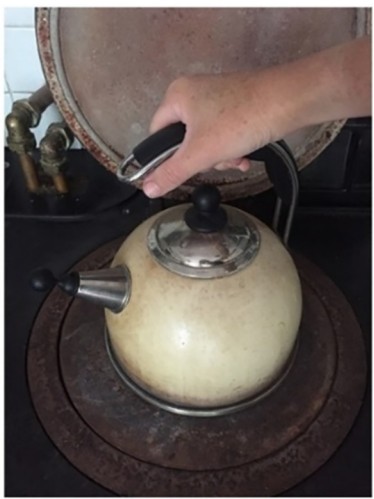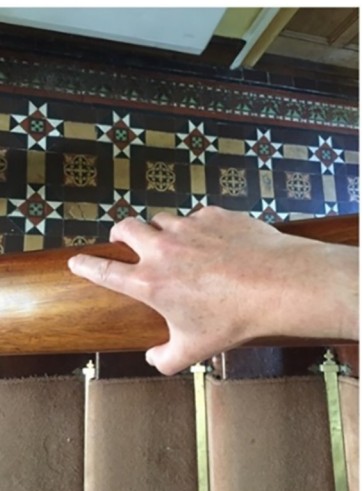

**Fig 9. How the penny dropped.** Morning weakness, achiness and stiffness in my hands meant not being able to open my fingers wide enough to hold the bannister on my way down stairs; it also meant struggling to properly grasp and lift the kettle as I made my morning coffee. These first symptoms were what lead to a RA diagnosis being suspected.

Indeed, a participant who encountered a physician who suspected she had gout rather than RA, described walking on her swollen foot to a follow-up medical appointment to prove the veracity of her diagnosis: *'I walked up so it would be swollen, sore, tender, all the things I didn't want [it] to be. To prove to [the physician] that I didn't have gout. That it was arthritis'*.

The photographer of Fig 11 reflects on his motivation to photograph his swelling and suggests it was borne out of a perception of not being believed, rather than any actual challenge to the legitimacy of his illness from others: *'It's not that I felt like I really needed to prove it to doctors, but you know they say that invisible illness thing, you always have that...your own insecurity. Do people believe me?'* Another participant who was concerned that people in her community did not believe she had RA reflected: *'Maybe it's me that thinks it, and not them'*.

This perception, or a feeling of needing to explain oneself to others, featured in many of the participant's experiences. For instance, explaining why they needed to walk more slowly, why they were wearing footwear not suited to a particular social occasion, or why they were absent from social activities:

There'd just been loads of injuries or just pain, so I'd always felt like I was nearly making it up...people thought I was making it up...because I always felt embarrassed nearly when I was saying I couldn't do things. I hated that. It was quite repetitive that I would be in pain...I shouldn't have cared, but it was just saying it to people.

As illustrated, in the context of an invisible illness, explaining oneself was an attempt to legitimatise reduced functioning or lack of participation and contribution.

**I don't want others to see my illness.** Participants attempted to avoid or minimise common stigmas associated with illness, such as being perceived as sick, old, lazy or weak. The stigma of RA as an "old person's disease" was described in either past or present experiences. At a point when her disease was not well treated, one participated recalled a troubling encounter:

It's like I was 80 when I was 40. That was really hard...I was walking down the road towards [home] one day and there were two young fellas there. [One] said [to the other]

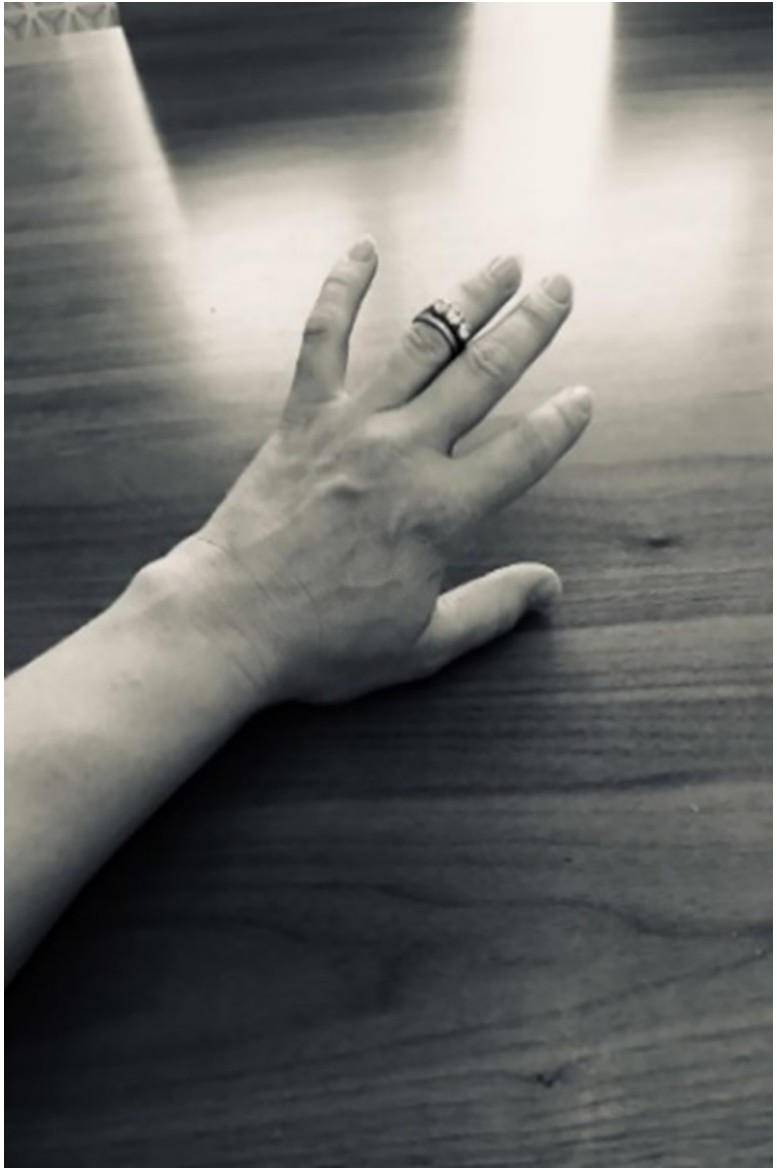

**Fig 10. Not today!** My rings not fitting almost became a representation of the beginning of my RA. They did not fit me for about 12–18 months after the onset of my RA symptoms. On the day of baby's christening I was upset that I was not wearing any rings so I bought a cheap ring; I love that ring so much now because I got comfort on that day that I was wearing a nice ring! My friend has a jewellery shop and she offered to change the size of my rings but I said no because that just meant to me I was accepting my swollen hands and rings were forever! . . . About a year after starting medication my fingers were OK and my rings fitted me again. But I have days now, they don't fit me and it just brings me back to the low period in my life where I was worried constantly about my fingers and hands. So this picture was taken on a bad day recently when I felt fatigued and my hands were swollen and my rings did not fit. It was like my hands were saying 'no rings today'! I just felt dreadful. I was really tired. My finger was really swollen; I couldn't put on my rings. And there's the picture. . . just like, not today!

"you're going to look like that when you're old too". And I wanted to say, I'm not old. I'm only 40. So that was very difficult, it's very difficult, when you talk about it. It brings up a lot of emotion really about how difficult it is or was. Because you try not to think about it too much.

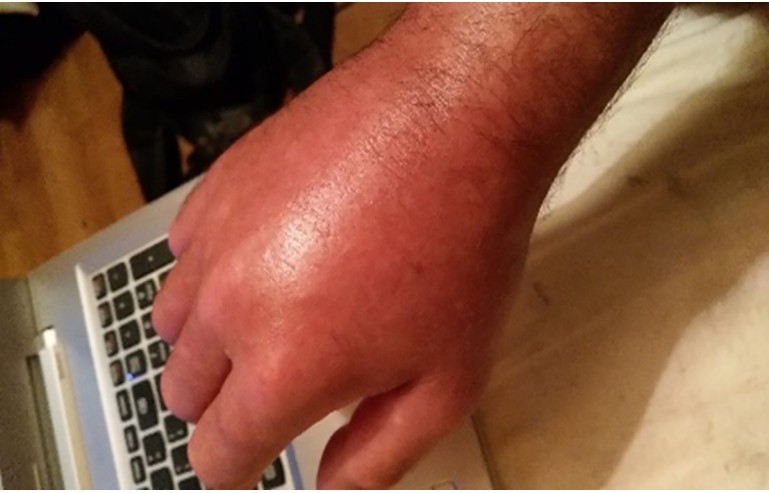

**Fig 11. Swollen hand.** I took that picture because a lot of people fail to understand if you say you're having a flare and they don't get how bad it can be. Most of the time when they see you, [it] would be when you're having a better day. So [its] like "oh, nothing wrong with him". So it's kind of handy to have that, I've actually got to pull that photo out a few times and say, you know, the days you don't see [me is] because my hands and my feet can look like that. . .Some people are like, "I get it", and it's actually been one of the most useful photographs to show a person when I'm having a flare up. . .And so this has been kind of a proof thing. . .but you know, it's stuff you can't see. But at least then they can visualize it. It's even going to the doctors as well, just so they can see it. Because they don't necessarily have swellings all the time and you've got pains and you can't.

As well as being perceived as old by others. Another described how she felt she was too young to live with chronic illness, reflecting a self-stigmatisation:

> I am too young to be lining up this many tablets, even if half of them are vitamins [and] supplements. I am too young to have to do this. I am too young to have to have to worry about managing a combination of meds. . .

In certain social contexts, managing illness could cause embarrassment. For example, avoiding the pain of a handshake, or lagging behind when walking with a group of people:

> If I'm with a group and I have to, we have to, walk somewhere, I dread it. Because I hate being the one that's going slower, so I try to keep up. But then you end up hurting and then you're also exerting so much more energy doing that. So, because for me it takes a lot of extra energy to push yourself that much forward. So, then you're worried, am I going to get sweaty? Because I feel like I've worked out during this [walk], but to everyone else it just feels normal. And it feels like, I don't know, I just dread that sometimes. Just walking places with people because I think, I don't know, you just feel weird.

Lagging behind others when walking was a social situation in which RA became visible and could evoke feelings of being a burden and holding people back, both literally and metaphorically. These kinds of social situations created an unwelcome visibility–or "surfacing"—of the illness. The pub, an important social space in Irish life, was a context in which social embarrassment could result due to a physical imbalance appearing as drunkenness. A participant described being in the pub and getting up to walk to the bathroom:

> It's very odd because if you go for a drink, you go in, you sit down, you have a drink, then you have to go to the loo. Then you go to stand up. People are looking at you because

you're hobbling. You're trying to get your balance [so] that you're not putting your weight on your bad joints. And then you get people, "oh look at her!" . . . [They think] that you've had more [to drink] than what they've seen you get.

Another described the same situation of getting up to walk to the bathroom in the pub. She expressed her concern around showing the illness and being perceived as severely disabled: '*I know they'd say "Oh my God*! *What's wrong with her*?*" because I have heard people saying that before*'. Fatigue and brain fog could also give the appearance of being drunk and disorientated, as described by another participant: '*when I go out shopping and the fatigue comes on—nearly always when I'm shopping—I've had girls say*, *you know*, *"She's drunk*! *Look at her"*.

Participants were keen to explain that they do not wish for sympathy or to be treated differently. However, they desired greater empathy and understanding of their illness. This was often reflected upon as a contradiction or paradox:

I don't really want. . .it's like if I had no hand or no leg people would see it. They'd go "Ah look, God love you. Get on the bus in front of me here" and all this. You don't want people to pick you out and say [something], but yet you still want them to know that you're not well. It's ridiculous, what I'm saying!

Another participant states: '*You want to fit in completely but you want people to understand what you are going through'.*

**I never get to forget about my illness.** Participants described the cognitive burden of living with and managing their illness. For many participants, self-management was constantly being negotiated and considered. The routine tasks involved in maintaining one's health and the associated cognitive load was invisible to others:

I would never tell people I'm going off for my bloods. I'd be meeting my friends and I'd say I'm going to get my bloods done. And they [would] say, "oh why you getting your bloods done" and I [would] say "for the rheumatoid arthritis" and they say "yeah, yeah I forgot about that" and I say "I never forget about it", you know. . .just that burden of always having to remember. It's the burden of it really, I suppose. Constant. And that's part of self-management.

They struggled to identify if increased pain, fatigue or physical limitations were caused by RA or simply a result of getting older and compared themselves with others as a frame of reference:

You're always getting older so, what is age and what is rheumatoid arthritis? It's always a question mark. I always try not to. . .put too much on it. But I'm wondering about fatigue. . .it's very hard. You can't ever know. So, I'm trying to be realistic but then wonder. [I] look at other people and see are they as tired?

Participants described how life with RA could not be spontaneous and the illness needed to be considered in much of their planning and decision-making; for example, ascertaining whether there would be stairs or a lift at any place they were visiting; making healthy food choices they could also prepare independently (i.e. chopping fruit and vegetables): '*I'm always trying to figure it out. . .I'm always trying to figure out what's going to make me better. It's just [I get] tired sometimes, it's all up here* [refers to head]. *So much up here. . .'.* The cognitive work of invisible illness could become mentally consuming and tiring.

## Medicine in all its forms

Photovoice is an action-oriented approach. While it was important to capture challenges within the lived experience of the group, it was also important to focus on what worked for this community and why. Participants offered many positive solutions to living with RA, to address practical, cognitive and emotional needs. Upon reviewing the photographs at the group workshop, the theme of *Medicine in all its forms* was identified.

While medical treatment and devices were helpful in everyday life, participants used the photo assignment to explore less obvious forms of support that they found helpful, such as creative expression, time spent in nature, and social support from family and friends. Fig 12 illustrates a child squeezing the participant's finger. This small physical gesture not only relieved pain for the participant but also provided emotional support. The participant derived comfort from his pain being acknowledged and seen by his grandchild, explaining '*she thinks of me. . .she sees it*'.

**I help myself.** Participants had accumulated a wealth of resources throughout their lives which they applied to the self-management of RA. This provided a degree of agency and control over the illness. Some participants were able to make use of their personal and professional networks to access treatment. A few participants were able to draw from their knowledge and experience of working in a healthcare setting. Others described practices they had used prior to their diagnosis that were helpful, such as swimming, yoga and time-management skills. Similarly drawing from experience with prior injury or illness could be helpful:

> I had a back problem many years ago. I learned not to walk around the stairs whereas once previously I would just run and go up the stairs. I learned to just take it nice [and easy] because just running up, things like that. It's all in your head".

Another participant remarked on learning to adapt one's activity: '*Pain is wonderful, it teaches you*'. Additionally, the cultivation of emotional resilience from previous experiences–particularly challenging ones–was described as useful in self-management. One of the

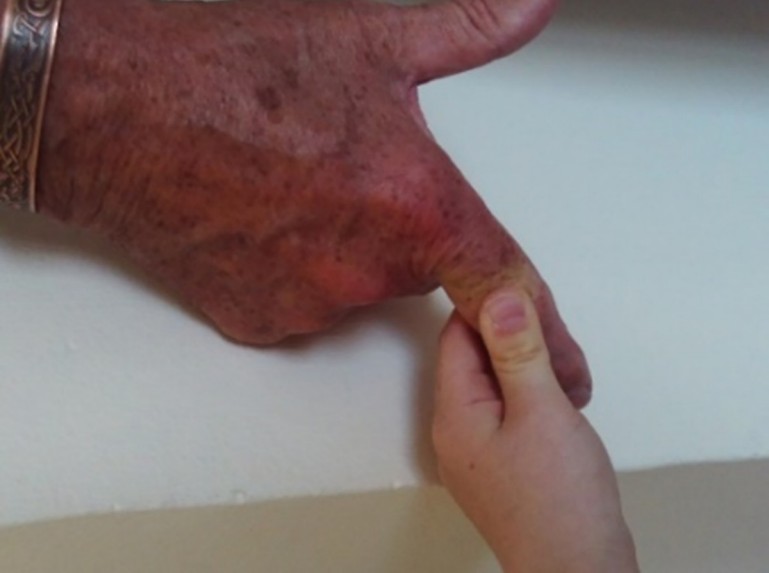

**Fig 12. Ah relief.** This is my granddaughter (aged 8), squeezing my finger as she realizes that by doing this, that it causes relief from pain for me. She does this voluntarily from time-to-time, especially when she sees me wince.

participants who lost her mother after a long illness, explained how that experience had shaped her attitude to illness:

> I've learned to compartmentalize. And with everything that we've been through in our family, you know, the stuff with my mom, I've just kind of, unhealthily learn to become numb in certain regards. . .you deal with so much, you just kind of have to become empathetic. But you do get a sense of numbness with everything. And you just kind of learn to push past it. I don't know how to describe it. It's kind of a strange feeling. . .when you've been through things before, you know that you'll get through it. And it's hard, but you'll eventually make it through.

**People help me.** Most participants described family as a significant source of emotional support. A participant described his wife as his "walking stick" explaining that she protects him:

> I've heard her in the kitchen and I could be somewhere, she says "Don't ask him to do such a thing because he won't be able to do it. Don't put him in that position that he has to refuse it". I've heard her say things like that.

Family could also provide practical and economic support. Those who did not have familial support mentioned close friends as a source of support.

In HCPs, empathy, patience and being informative were characteristics valued by participants. A participant describes her experience of being putting on methotrexate for the first time:

> My rheumatologist, he sat down and told me the history of the medicine. He was like "it started out as this", and he explained everything to me. He was so explanatory and he would just touch my joint and he'd be like "Oh, you hurt, don't you?" And just so kind. And any time I had any questions, he took the time with me and explained. He was the only practitioner in [his] office and he just genuinely cared.

An absence or lack of empathy was also experienced. For example, when asked whether family play a role in supporting her, a participant remarked, *'Ah no, they just laugh at me when I tell them not to walk on my feet'*. She goes on to elaborate on the problematic nature of the invisibility of pain, and how deformity of the feet presents an inconspicuous disability:

> I don't think they understand what it is like. When [a family member] saw the photographs of the toes, she said "oh I didn't know they were that bad" but people don't tend to look at your feet. They really don't. So, it's not that they don't understand, [it's] that they don't understand the pain. You can't explain the pain that you have with the arthritis. Like nobody understands that, unless they get it. And you sort of hope they don't ever get it.

**Things help me.** Many participants mentioned using devices to overcome obstacles posed by immobility, stiffness, a lack of strength or dexterity in the hands. There were commercially available devices such as specialized can and bottle openers. But there was also a cognitive—and sometimes creative—task of identifying what items could help the person, such as pre-chopped vegetables, or making modifications and adaptions to household items such as handles, taps, washing lines, chairs, and toilets.

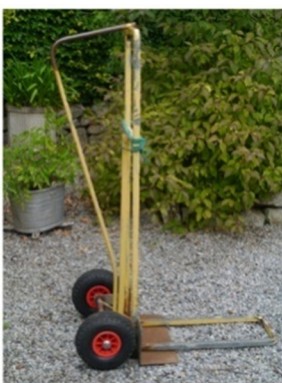
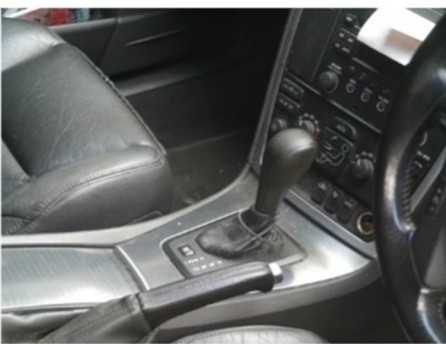

**Fig 13. Achievement without pain.**

This task of finding things that worked was described as a way of outsmarting the illness (as shown in Fig 13); again, echoing the need for increased agency over RA. When discussing this theme during the workshop one of the participant's explained:

It's almost like playing hide and seek with your disease, you're trying to get the best of it every day. . .And then find ways to out-fox it and out-smart it, and that's what you're doing, you know, with changing your car and finding ways to lift things.

Devices and aids, such as shower stools, walking sticks and frames could provide necessary support at times of reduced mobility, however they could be stigmatising, especially for younger people (Fig 14).

Thus, accepting these devices provoked complex emotions. One participant in her thirties reported struggling with medication compliance but was reluctant to use a pillbox: *'I associate that [pill box] with my granny when she was very sick, so I don't want a pill box. I don't want that. That would be like I have an illness. So, I have to get my head around that'.*

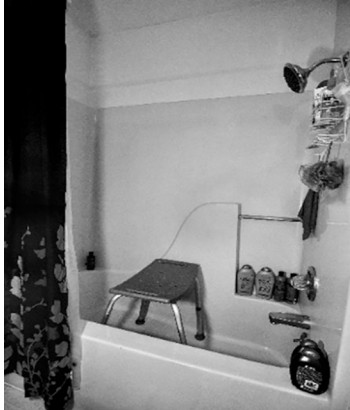
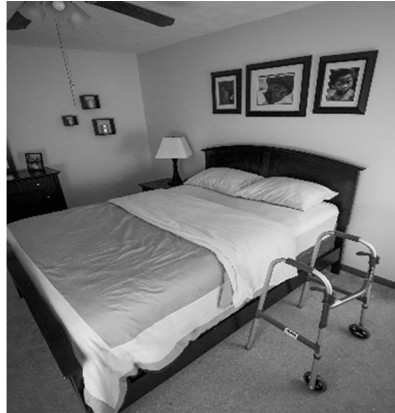

**Fig 14. Walker and shower.** This is my old bedroom, and this is the walker that I had to use after my surgery. . . Because you think of walkers as [something that] people that are in their eighties or nineties use. And I know it was for a surgery, but. . . it's less about the walker, but showing what you have to go through at such a young age sometimes when you have this. So, I didn't have to use it terribly long, maybe a few weeks, . . . it also represents how old you feel in your body as well.

Participants described how medication and treatment were important to their health, well-being and functioning. One participant remarked that she had come '*back from the dead*' following successful treatment. However, the journey to find effective treatment was often reported as a long and difficult process:

> One-time last year I went in and actually saw my doctor, my specialist and I said "look, thanks a million for the 10 years [of treating me, but] I've no life. . .I can walk, and I can move and I'm alright. But I don't have a life". Methotrexate kills off that and I had checked with people it wasn't just me. He says "We can't have that". He says "Have you got money to pay for an injection?". I said "no". He said "well I'll put in a case for you" and he changed my medication.

Once effective medication was found, there was a profound fear among some participants of changing treatment. For example, participants worried about an increased financial burden of new medications, or the disruption posed by side-effects:

> I just don't do medication very well. Any medication I've had to take so far it takes me weeks to get. I get all the side effects; nausea, headaches. I get them all double whammy. Even the [names drug], is very well tolerated by people and it took me 4 months of severe, I'm talking like banging hangover, morning, noon and night so that would be my fear. Big fear. Then trying to function with that.

Another participant was in the process of changing medication for the fifth time in four years. Most recently, she had been hospitalised having experienced complications following immunosuppressant treatment. Yet she reflected that her greatest fear would be to run out of treatment options:

> My biggest fear [is] that I'll run out of options and be forced to go on something like methotrexate or one of the older, less sophisticated drugs, because I really don't want to have to do that if I can. So, I'm glad that there are still options there, within more high-tech, more sophisticated meds. I knew the ones that I was on really weren't working. They were making me very sick. I've been sick five times inside six or seven weeks or taking them.

## Mind yourself

The theme "*Mind Yourself*" was another solution-oriented concept generated by the participants. Broadly, it attempted to capture forms of self-care, particularly activities that were perceived to be psychologically therapeutic, as refelcted in Fig 15.

In preparing for the exhibition, only one photo was assigned to this theme, however the idea was frequently expressed throughout the interviews.

**I fight RA mentally.**   RA presented cognitive and emotional challenges. Optimism and positivity were identified as an important tool in managing RA. For example, a participant described how she honed the skills to pace herself and develop a positive mental attitude:

> I'm always at 100%. I mean before I was diagnosed with this, I'd be [busy with] work, my parents, the kids, everything. Whereas now, I would have that skill of saying, listen make your[self] a priority now. . . Make a plan [of] how are you going to get through today because tomorrow is a new day and you wake up feeling much better. . .Before, with the medication, I would be so dreadfully fearful that I would have this forever. That I would

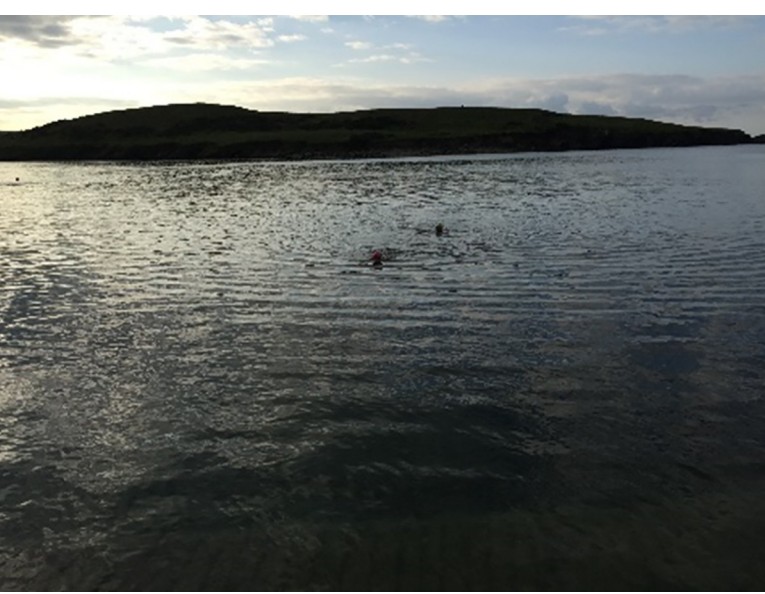

**Fig 15. Minding mind and body.** Minding mind and body, one splash at a time. Early morning swims with my local swimmers group helps me feel strong physically and mentally. Out at sea, whilst moving through the waves, listening to the seagulls and watching out for jellyfish; I'm taken away from terrestrial problems of aches and pains, and wonderfully I get to carry an element of this buoyancy through the day.

never feel ok again. That this illness is part of living now. And while it is part of it now, I actually [have] the skill now to say it's only a few days; it's only a few hours; its one day and tomorrow will be a new day. And you get up and start all over again. I think you have to have a very, very strong line in your head where you have to say. . .don't be dragged down that route where it's all awful and it's all pain because it's not. If you give yourself. . .let yourself be taught the skills.

Fear was an emotion that needed to be managed as part of the illness. The fears cited by participants included experiencing flare, disclosing illness at work, side-effects of medication, severe immobility, increased dependency on family, and a fear of the physical pain of being accidently injured by others. For example, a participant explained how she was often bumped into by shopping trolleys:

They [shoppers] bang your ankle and [say] "oh sorry" and you're [thinking], oh my God, I'm going to be so sore now for the rest of the night. Little things like that. But you can't turn around and say, "Come here you, I have arthritis!" What good is it going to do? It's not going to make the bang or the bump any better. You might feel better for letting off a bit of steam but you're still going to be sore.

This participant's experience reflects the problematic nature of an unseen illness in society, as others cannot appreciate the level of pain incurred by a simple accident (e.g. bang or bump). Even if the illness is disclosed, it does not minimise the pain experienced. This participant explained that she has adapted her behaviour and now visits the supermarket in early mornings when it is less busy. Another useful self-management tool described by participants was the use of downward social comparison whereby people compared themselves with others who were less well off. A participant who had worked with young people with severe

disabilities and described being inspired by their achievements, explained: '*I would have thought that they were worse off than I am*'.

**Things I do to look after myself.**   Participants described how they took care of their mind and body as part of managing their illness. Socialising with friends, swimming, writing and being creative were all therapeutic forms of play alleviating the cognitive load of illness. As one participant talks about his woodwork hobby explains: '*It takes you out of yourself, you forget*'. Another participant who routinely swam in the sea explains: '*I get to forget, you know my body, rheumatoid arthritis doesn't exist*'. Part of self-management and effective self-care was learning to recognise one's limitations. Participants spoke about learning to trust that things won't all fall apart, to give themselves permission to rest if they need to–a particular concern expressed by mothers of young children. A mother of three states:

> Mentally then, I guess, it's difficult to accept that you have a lifelong condition and try not let it be you or define you. . .I acknowledge that it's there rather than accept that it's there you know. You work your way around it and you have to adjust your limitations and decide what are the things that really matter as opposed to *have to* matter.

Conversely some participants explained that they would not be limited by RA; that they would not "give in" to it. Maintaining independence and having agency over the illness was a constant negotiation.

## Discussion

Our results shed light on the lived experience of RA from the person's perspective, as well as providing insight into the value of photovoice. The participant's experiences and reflections on managing their illness is demonstrated in their images and words, which can inform future care and support for this population. The relationship between agency, participation and contribution was central to the lived experience of RA. Participants expressed a need to develop greater agency over the illness, largely as a means of increasing their participation and contribution to family, work or society thereby reducing feelings of being a burden or dependent on others.

Armstrong [79] argues that agency, as applied to patients and their behaviour, has been the product of a fundamental reconstruction of patients' identify in the latter half of the 20th century, shifting from the concept of patients as passive, docile subjects to a new vision of patients increasingly responsible for their own treatment and actions; who share decision-making with their doctors; and actively voice their inner life-worlds. However, as this study demonstrates life with RA often meant accepting reduced agency, to an extent that some participants described the disease as being in control. This is consistent with findings from previous studies of RA patients whereby higher disability and pain rates are associated with lower self-agency [80]. Participants either integrated RA into their identity or considered it as something external to, and distinct from, the self [81]. Our findings suggest that participants continue to struggle to voice their inner life-worlds to many people important to their physical and emotional care, such as family, friends and colleagues. Thus, the ideal of the actualised patient may be far from realised. Furthermore, we propose critical reflection on these norms and expectations within contemporary culture observing that participants battled with a sense that they should be "doing more" and that life was going on around them but they could not take part.

The visibility and invisibility of living with RA were interchangeably problematic for participants and some reflected on this as a contradiction or paradox; yet we propose this should be conceptualised along a continuum of surfacing and masking of illness in an effort to maintain

"normality" and avoid the stigma of disability and otherness. Because many of the symptoms of RA are hidden, participants described situations or contexts where they had to decide whether or not to disclose their illness, for example, to romantic partners or a potential employer. Disclosing illness and disability has been examined in previous studies, particularly in the context of the workplace [82, 83]. However, we expand upon the notion of disclosure to propose a "surfacing" of RA. As illustrated, participants expressed moments where the progression and activity of the disease surfaced either to themselves or others. Where individuals had agency over this surfacing of RA, it could improve the understanding and legitimacy of their illness for others and indeed, for themselves. Conversely, in situations where the participant had limited agency, the surfacing of RA was viewed as an unwelcome reminder of their illness and could represent a disconnect between the body and self [84]. The concept of "passing for normal" to ensure that no one knows about the condition—is a recognised behaviour in chronic illness, and of particular concern with highly stigmatised conditions such as HIV [85, 86] and mental illness [87]. Among a RA population, our findings suggest individuals were constantly negotiating a balance between invisible "normality" and visible disability and "otherness". Flares in symptoms such as pain and fatigue, could tip individuals over from one side to another. Participants described instances where they attempted to "mask" their illness in an effort to minimize stigma [88] and adhere to social norms. However, these efforts could fail when overpowered by increased disease activity. While some studies have addressed masking of deformity in RA [89, 90], a broader exploration remains underdeveloped.

With some exception, exploration of the relationship between visibility and invisibility is spare, lacks cultural analysis and tends to be limited to conditions such as Irritable Bowel Disease and Multiple Sclerosis (MS) [91–94]. From a meta-study of qualitative and quantitative research of the lived experience of chronic illness, Joachim and Acorn [94] developed Goffman's [88] seminal work on stigma to propose a theoretical framework of stigma and factors that influence disclosure or non-disclosure of chronic illness. However, they did not discuss the combination of visibility and invisibility within a disease trajectory nor the experience of fluctuating between these states. Hoppe [93] explores visibility and invisibility among people living with MS in Dublin, Ireland conceived as agents strategically enacting the visibility of their illness in various settings. While this research recognises that people move between categories of visible and invisible illness and considers cultural context, unlike our findings, it affords the individual a high level of agency. Thus, we propose further examination within chronic illnesses of the interplay between visibility and invisibility and the role of agency.

A substantial body of work exists in relation to social stigma; a concept that may explain factors that underpin the (non)disclosure of RA. Participants in our study described attempts to conceal their illness in situations where stigmatization could occur from others, but also expressed a desire to "not see" their own illness at times, reflecting forms of self-stigmatization or self-devaluation. Again, drawing on Goffman [88], Jones et al. [95] developed dimensions of stigma as "concealability", the degree to which the condition is hidden or visible; and *"controllability" referring to a* belief that the cause or origin of the illness lies with the individual and if they cannot get better on their own, they lack personal effort. These ideas fit with the long established theory of illness as a social construct [96, 97] suggesting that 'illness is essentially social since it refers to undesirable deviation from accepted norms of health and appropriate behaviour' [97, p.176]. Arguably, the increasing availability of chronic disease self-management educational programmes [22] and the rise of the self-care movement, may contribute to the shaming of the chronically ill individual who is unable to "get better". We suggest further investigation of the relationship between masking and surfacing as forms of concealing and revealing chronic illness.

This study sought to identify solutions to living with and managing RA. The participants identified the themes, "Medicine in all its forms" and "Mind Yourself" to capture what worked in terms of self-managing RA. Part of self-management involved participants advocating for better treatment and a better quality of life, ideally this was done in collaboration with their HCP. However, finding the right medication was clearly a lengthy process that required resilience, determination and optimism. Participants focused on solutions that supported them to manage changes for example, in their medication, treatment, functionality and increased dependency. As the condition unfolded, finding what worked, as well as creative ways of adapting, was part of the learning curve of self-management. Minding oneself often involved activities that provided an opportunity to unload the cognitive burden of illness and its management–allowing oneself to forget. Physical relief from these activities was expressed as being secondary to the cognitive and emotional benefit.

## Strengths & limitations

To the best of our knowledge, this is the first academic study of rheumatoid arthritis using the photovoice method [36]. Photovoice proved to be an effective tool for actively engaging individuals in reflecting and communicating the lived experience of self-managing RA as evidenced in the richness of the data presented here. Moreover, there was added value of using photographic images above and beyond narrated experiences that may be difficult to disclose and articulate in verbal or written form [37–40]. Anecdotally, participants explained that they found the process of personal reflection, and the sharing of mutual concerns and experiences rewarding. The participants echoed a sense of empowerment from their participation, as has been demonstrated by using this method [98]. Participants engaged positively with the photo assignment. Following the introductory workshop, additional instruction on the method was not required, and this small sample of people with RA produced rich qualitative data. Although basic digital cameras were gifted to all study participants, most (n = 7) chose to use their own cameras or smartphones to take photos. In some instances, the basic digital cameras proved challenging to use; most likely attributed to either faulty devices or the difficulties in operating the camera and saving photos. Participants tended to avoid taking photos that identified people. Only one participant included an identifiable subject in their photos. This photographic subject signed and submitted an appropriate release form.

It should be noted that the photovoice method requires sustained commitment from highly-motivated participants and the researcher. Indeed, this was reflected in our recruitment phase whereby public calls for study volunteers were three times more successful than recruitment from a hospital rheumatology clinic. Arguably, people who were motivated to actively make contact with the researcher to enrol, were more likely to be retained in the study—some even travelled large distances such was their motivation to participate. This ensured we had an active and engaged study group but presents a possible limitation in that our sample might have a higher general interest in concerns regarding RA policies and science, in comparison to others. A funded fellowship supported this project and meant that the lead researcher (SD) was highly involved in all aspects of the study and ensured that good relationships were built with participants. The strength of these relationships and the consistency of personal contact with the lead researcher throughout was arguably a key component of the project's success.

We endeavoured to involve participants in generating and analysing data, as well as reviewing results of analysis, and a draft of this manuscript.

While a high level of involvement was observed throughout the fieldwork period, responses were relatively low when the researcher **[SD]** contacted participants (approximately six months later) to review a synthesis of analysis (Fig 2) and a draft of this manuscript. Four

participants contacted the researcher [SD] to confirm that the themes, as shown in Fig 2, were valid. All participants were invited to read a draft of this manuscript but only three requested a copy, and their feedback was limited to confirming they were happy with it. In-depth face-to-face member checking procedures, such as interviews or focus groups, might have generated greater involvement [63], or an explicit invitation to contribute directly to the manuscript [99]. As a form of participatory action research, photovoice is orientated towards pursuing practical solutions, as well as the flourishing of individuals and communities [100]. The action orientation of this study aimed to effect change for this community by identifying policy changes and practical solutions for self-managing RA. However, as the study evolved it became clear that the first task was to meaningfully capture and communicate the lived experience of self-managing RA, and involvement of policymakers–and other stakeholders–became less of a priority. This may have limited the impact of the study. However, the research process in itself was powerful as a tool to facilitate reflection for participants and, to some extent, was the unanticipated outcome of the project. The blurring of boundaries between photovoice as a research tool and therapeutic intervention has been observed by researchers [101–104], yet it often remains covert in research design and practice. While researchers may be wary of lengthy and bureaucratic ethical procedures associated with the explicit use of photovoice as an arts-based therapeutic intervention, we encourage greater recognition and evaluation of the method in this regard. The ability of photovoice to aid disclosure and discussion of experiences, improve data and provide health benefits should be harnessed [105–107].

## Conclusions

This study contributes to the literature on the lived experience of rheumatoid arthritis, the self-management of chronic and invisible illnesses; and a growing body of work in photovoice research. Through a reflexive process of individual interviews, photographic exercises and group discussions, participants expressed the lived experience of self-managing RA, highlighting the experience of reduced agency, contribution and participation; varying states and circumstances around the masking and surfacing of the illness; and the ways in which they found psychological and emotional resilience. Our findings suggest that RA is experienced as a fluid relationship between states of masking and surfacing shaped by contextual and situational factors. The extent to which the individual has agency to control the visibility of their illness across various social and environmental contexts–particularly where the individual feels a heightened expectation that they *should* be better able to control their illness—is arguably a crucial component of self-management. As RA is an illness producing both visible and invisible disabilities, this is especially pertinent. These results can inform the development of interventions for conditions, such as RA, that support individuals to have greater agency over how they manage both visible and invisible aspects of illness and disability. The series of public exhibitions that were held as part of this study have increased public awareness and contributed to educating the public about on the unseen challenges of living with a chronic illness.

## Supporting information

**S1 Appendix. Participant information leaflet and consent form.**
(DOCX)

**S2 Appendix. Overview of group workshop 1.**
(DOCX)

**S3 Appendix. Overview of group workshop 2.**
(DOCX)

**S4 Appendix. Phases of thematic analysis.**
(DOCX)

**S5 Appendix. COREQ checklist.**
(DOCX)

## Acknowledgments

We would like to thank all members of our study group and research advisory group who generously gave of their time and energy and openly shared their thoughts and personal experiences. Many thanks to *Arthritis Ireland*, Patricia Kavanagh and colleagues at the Mater Hospital, and Dr Emma Dorris and the Patient Voice in Arthritis Research for their support throughout the project.

## Author Contributions

**Conceptualization:** Susie Donnelly, Anthony G. Wilson, Hasheem Mannan, Thilo Kroll.

**Data curation:** Susie Donnelly, Laura Whitehill.

**Formal analysis:** Susie Donnelly, Hasheem Mannan, Laura Whitehill, Thilo Kroll.

**Funding acquisition:** Susie Donnelly.

**Investigation:** Susie Donnelly, Anthony G. Wilson, Claire Dix.

**Methodology:** Susie Donnelly, Hasheem Mannan, Thilo Kroll.

**Project administration:** Susie Donnelly.

**Resources:** Susie Donnelly.

**Supervision:** Anthony G. Wilson, Hasheem Mannan, Thilo Kroll.

**Validation:** Susie Donnelly, Hasheem Mannan, Thilo Kroll.

**Visualization:** Susie Donnelly, Claire Dix, Laura Whitehill.

**Writing – original draft:** Susie Donnelly.

**Writing – review & editing:** Susie Donnelly, Anthony G. Wilson, Hasheem Mannan, Claire Dix, Laura Whitehill, Thilo Kroll.

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
