## [Decision Letter · Decision Letter 0]

6 Nov 2020

PONE-D-20-20019

(In)Visible Illness: A Photovoice Study of the Lived Experience of Self-Managing Rheumatoid Arthritis

PLOS ONE

Dear Dr. Donnelly,

Thank you for submitting your manuscript to PLOS ONE. After careful consideration, we feel that it has merit but does not fully meet PLOS ONE’s publication criteria as it currently stands. Therefore, we invite you to submit a revised version of the manuscript that addresses the points raised during the review process.

Thank you for submitting this interesting manuscript. You will see that both reviewers have provided favourable feedback and also identified a number of areas for improvement, e.g. in terms of highlighting the gap in the literature that you are contributing to and making sure your statements are supported by appropriate evidence. They have also requested more detail on your methods. I disagree with the suggestion from reviewer 1 that quotes would be better located in a table; I think your findings section is well presented. Feel free to defend the current layout in your rebuttal letter. Otherwise, all comments need to be addressed and should help to improve the quality and readability of the manuscript.

We look forward to receiving your revised manuscript.

Kind regards,

Shelina Visram, PhD, MPH, BA

Academic Editor

PLOS ONE

Journal Requirements:

'Funding for this research was awarded to SD under a Wellcome Trust Institutional Strategic Support Fund which was financed jointly by University College Dublin and the SFI-HRB-Wellcome Trust Biomedical Research Partnership as part of a Medical Humanities and Social Science Collaboration Scheme (Grant number 204844/Z/16/Z).

www.wellcome.ac.uk

The funders had no role in study design, data collection and analysis, decision to publish, or preparation of the manuscript.'

We note that one or more of the authors are employed by a commercial company: Independent filmmaker, Dublin

4. Please ensure that you refer to Figures 1, 7, 13 and 15 in your text as, if accepted, production will need this reference to link the reader to each figure.

5. Please include captions for your Supporting Information files at the end of your manuscript, and update any in-text citations to match accordingly. Please see our Supporting Information guidelines for more information: http://journals.plos.org/plosone/s/supporting-information

Reviewers' comments:

Reviewer's Responses to Questions

**Comments to the Author**

1. Is the manuscript technically sound, and do the data support the conclusions?

Reviewer #1: Yes

Reviewer #2: Yes

2. Has the statistical analysis been performed appropriately and rigorously? 

Reviewer #1: N/A

Reviewer #2: N/A

3. Have the authors made all data underlying the findings in their manuscript fully available?

Reviewer #1: Yes

Reviewer #2: Yes

4. Is the manuscript presented in an intelligible fashion and written in standard English?

Reviewer #1: Yes

Reviewer #2: Yes

5. Review Comments to the Author

Reviewer #1: The authors have done an excellent job describing the lived experience of people living with RA through a combination of interviews, focus groups, and photos. I also appreciated how closely patients worked with the research team and were considered key stakeholders in the analysis and interpretation. I do think the manuscript could benefit from some editing, and I have a few questions:

For sample selection, was participant race considered?

How was it decided which group participants would be assigned for the workshops?

Was the RAG involved in the final analysis or did only participants provide feedback?

I appreciate the inclusion of multiple quotes to underscore many of the authors’ main points. I wonder if a table displaying these quotes together would be beneficial to keep from going from text to quote several times during the results. Overall, I think the results could use some editing to highlight the key findings without including long quotes for each section.

Did you find that 11 participants were enough to understand the full spectrum of experiences? Do you think thematic saturation was achieved?

Reviewer #2: Summary of the study and general comments

Thank you for the opportunity to review this manuscript on a topic that is timely and relevant.

This paper presents the results of a qualitative study conducted in Ireland which explored the lived experience of self-managing rheumatoid arthritis.

Overall, this is a well-conducted study that makes important contributions to the research field. Suggestions on how to strengthen this paper follow.

Abstract

Clear and structured abstract. It includes a detailed description of the study, participants, what was done and what participants found.

Introduction

Clear and nicely written Introduction, which sets well the context and background of this study. The importance of rheumatoid arthritis is clearly explained.

Lines 92-93 – you mentioned previous qualitative studies. Please summarise them briefly, highlighting any strength or limitation. It is important to highlight any gaps in the literature before moving on to present photovoice.

Lines 95-97 – please provide references for this statement. Previous authors have highlighted that one of the strengths of using photovoice is the combination of various methods, and how this may help us in accessing people’s perceptions in a deeper way than solely relying on interviews/group discussions. Some examples are below:

E. D. Carlson, J. Engebretson, and R. M. Chamberlain, “Photovoice as a social process of critical consciousness.,” Qual. Health Res., vol. 16, no. 6, pp. 836–52, 2006.

S. Ronzi, D. Pope, L. Orton, and N. Bruce, “Using photovoice methods to explore older people’s perceptions of respect and social inclusion in cities: Opportunities, challenges and solutions,” SSM - Popul. Heal., vol. 2, pp. 732–744, 2016.

L. Liebenberg, “Thinking Critically About Photovoice,” Int. J. Qual. Methods, vol. 17, no. 1, p. 160940691875763, 2018.

Lines 103-106 – please provide references for these statements,

Lines 106-107 why is photovoice especially well-suited to the study of invisible illness as a means of exposing the everyday realities and hidden conditions of people’s experiences? If you include this statement right at the start, you should support this by evidence from previous literature. Also, this statement does not fit well your description of the method (it disrupts the flow). I suggest you move it later in the paragraph (and provide justification for this statement).

Lines 111-113 –whilst it is ok to use quotes, try not to overuse them and rephrase what you want to say in your own words.

Methods

Line 138 – do you have any reference that you can include of past photovoice work?

Lines 152-153– please define what you mean by Chronic Disease self-management Programme

Lines 153-154 what are the components of a formal self-management education programme in this context? Please provide further details and support these with references.

Lines 154-156 – this sentence reads a bit complex to follow for a reader who is outside your immediate field. Please simplify this sentence by providing further details (especially define ‘formal self-management discourse’ in this context).

Lines 158-159 why did you use two recruitment strategies – clinical and public – to simultaneously obtain a sample of eleven participants? What was the rationale? What was the added benefit of doing this? Please expand the rationale for this choice.

Procedure:

Lines 172-174 – who led the fieldwork and recruitment?

To add transparency, please list the initials of the research team involved in discussing emerging themes and reviewing fieldwork also earlier on (in addition to line 247).

Line 181 –what was the rationale behind having each workshop of three hours in duration?

Lines 187-188 – were examples of photos provided to participants? Did any participant ask for further explanation of the photo task? If so, what did you say?

What ethical issues were discussed?

Use of consent forms for photographic subjects: what did the consent form include? What ethical guidance did you refer to and have used in this photovoice study? Please reference any guidance accordingly.

Was the camera left to participants for 2 weeks?

Was there any follow-up conducted with participants during this time?

Did any of the participants encounter any challenges in taking photos during the 2 weeks? If so, what was and how did you resolve it?

Why did you the decide to divide participants into groups of two to three (workshop 2)?

Lines 215-217 “Group B were asked to review photos following the same procedure as Group A, but at the end of the workshop they were presented with Group A’s themes to incorporate into their selection”. How did this work in practice? Please expand.

Lines 226 - it is great to see the photo-exhibition organised. How many people attended the event? And who were the attendees? (e.g. policy makers, community members…) How many participants attended and presented their photos at the event?

Were all participants involved in the analysis? Fig. 1 and Appendix should be explained also in the text, to assist the reader in understanding what was done.

In particular, explain how participants were involved in coding the photos and identify main themes, and how you have linked this to your analysis. How did you incorporate participants’ coding and identification of main themes in the thematic analysis that you conducted? It would be valuable to understand how these two elements were linked. This will support what covered in Appendix S2 and S3.

The appendices the authors provide are very informative and detailed and add strength to the overall paper. However, I suggest to expand this section (see above questions), as it is important to present how photovoice methods were applied in this study.

Line 256 – please change the verb into the past tense

Very clear explanation on how Public and Patient Involvement was undertaken in this study.

Results

Overall, the Results section includes some fascinating findings. Suggestions on how to improve this section follow.

Figure 3 – can you provide a quote supporting the explanation of this photograph? This comment applies to all photographs that don’t have a supporting quote. Although the authors explain the reported meanings associated with each photo (and in some cases, a supporting quote is presented), for transparency, each photo should be presented with the participant’s explanation as well.

Also, next to each quote, I recommend you include the participant number, gender, and age. Having this info helps the reader to contextualise the findings. E.g. (PA, F, 37). Otherwise, the quotes feel a bit difficult to follow and put into context.

The same applies for the photos presented. Please specify which participant took each photo.

Line 628 – what is HCPs?

Discussion

Clear and detailed discussion of the findings in the context of the literature.

Strengths and limitations:

I suggest the authors to support many of their statements with appropriate references.

Lines 870-872 – Please add how photovoice proved to be an effective tool for actively engaging individuals in reflecting and communicating the lived experience of self-managing RA. What did you observe? What did you find?

Lines 872-873 – please add why you felt that this was the case (do you have any evidence of this? What did you observe?). It would be good to comment further and expand this statement: “participants reported to find the process of personal reflection, and the sharing of mutual concerns and experiences rewarding”. As suggested above, there are some papers that have examined and assessed the value of using a combination of photos, interviews, and group discussions to access people’s lived experiences/perceptions. Please refer to examples made in Lines 95-97.

Overall, a well-structured discussion and presentation of the strengths and limitations of this study.

6. PLOS authors have the option to publish the peer review history of their article (what does this mean?). If published, this will include your full peer review and any attached files.

Reviewer #1: No

Reviewer #2: No

---

## [Author Response · Author response to Decision Letter 0]

30 Dec 2020

Editor 

1 Please ensure that your manuscript meets PLOS ONE's style requirements, including those for file naming.

Done - The authors believe the manuscript meets the journal’s style requirements. We would be grateful if you could please advise if we do not, and where specifically. 

2 We note that you have stated that you will provide repository information for your data at acceptance. Should your manuscript be accepted for publication, we will hold it until you provide the relevant accession numbers or DOIs necessary to access your data. If you wish to make changes to your Data Availability statement, please describe these changes in your cover letter and we will update your Data Availability statement to reflect the information you provide. 

Done - Citation to archived dataset (with DOI) inserted in-text, and updated in data availability statement. 

3 Financial Disclosure section:

Done - updated and submitted with cover letter and above.

4 Please ensure that you refer to Figures 1, 7, 13 and 15 in your text as, if accepted, production will need this reference to link the reader to each figure. 

Done - These figures are now been referred to in-text.

5 Please include captions for your Supporting Information files at the end of your manuscript, and update any in-text citations to match accordingly. Please see our Supporting Information guidelines for more information: http://journals.plos.org/plosone/s/supporting-information

Done -A supporting Information section has been added at the end of the manuscript. It lists all of the appendices with titles and caption, and this has been updated in-text where the appendices are referred to.

Reviewer 1 

1 For sample selection, was participant race considered? 

N/A Did not include race as an explicit selection criterion. Study was conducted within an ethnically homogenous population (White Irish)

2 How was it decided which group participants would be assigned for the workshops? 

Done This was a random selection. Added to text.

3 Was the RAG involved in the final analysis or did only participants provide feedback? 

Done - Due to limitations imposed by Covid-19, the RAG was not involved in the final analysis. The study team had hoped to hold a workshop with the RAG and study participants. Reduced time and resources during lockdown mean that this workshop was not held. This is now explained in the Data Analysis section.

4 I appreciate the inclusion of multiple quotes to underscore many of the authors’ main points. I wonder if a table displaying these quotes together would be beneficial to keep from going from text to quote several times during the results. Overall, I think the results could use some editing to highlight the key findings without including long quotes for each section. N/A - The authors thank Reviewer 1 for this suggestion. However, on reflection the team believes the findings are well presented and provide an important first-person narrative which grounds the findings section.

5 Did you find that 11 participants were enough to understand the full spectrum of experiences? Do you think thematic saturation was achieved? 

Done - Thank you for raising this question. The authors contend that data saturation was reached. This is now stated in the Findings. 

Reviewer 2 

1 Lines 92-93 – you mentioned previous qualitative studies. Please summarize them briefly, highlighting any strength or limitation. It is important to highlight any gaps in the literature before moving on to present photovoice. 

Done - Thank you for highlighting. This has been briefly addressed with a summary of some relevant studies cited and where gaps exist. 

2 Lines 95-97 – please provide references for this statement. Previous authors have highlighted that one of the strengths of using photovoice is the combination of various methods, and how this may help us in accessing people’s perceptions in a deeper way than solely relying on interviews/group discussions. 

Done - Thank you for these references. They have been integrated into the paper, and additional literature on photo elicitation in interviews has been added to support claims.

3 Lines 103-106 – please provide references for these statements, 

Done - Citations added.

4 Lines 106-107 why is photovoice especially well-suited to the study of invisible illness as a means of exposing the everyday realities and hidden conditions of people’s experiences? If you include this statement right at the start, you should support this by evidence from previous literature. Also, this statement does not fit well your description of the method (it disrupts the flow). I suggest you move it later in the paragraph (and provide justification for this statement). 

Done - Thank you for raising this. This sentence has been moved to the bottom of the paragraph and additional citations included.

5 Lines 111-113 –whilst it is ok to use quotes, try not to overuse them and rephrase what you want to say in your own words. 

N/A Thank you for the suggestion. One of the in-text quotes from Conrad and Baker has been rephrased in our own words. However, on reflection the authors wish to retain this particular quote as it is a core definition from the originators of the methodology (Wang & Burris) and we believe it is valuable to preserve its intactness. 

Methods 

6 Line 138 – do you have any reference that you can include of past photovoice work? 

N/A Unfortunately, this is the first published photovoice study for the authors. The first author has previously used photovoice in a commercial setting as part of research and development for a medical device. However, the study was commercially sensitive and therefore not published. 

7 Lines 152-153– please define what you mean by Chronic Disease self-management Programme 

Done An explanation/definition of CDSMP and self-management programs as well as their limitations are now described in the Introduction. 

8 Lines 153-154 what are the components of a formal self-management education programme in this context? Please provide further details and support these with references. Done As above. See Introduction. 

9 Lines 154-156 – this sentence reads a bit complex to follow for a reader who is outside your immediate field. Please simplify this sentence by providing further details (especially define ‘formal self-management discourse’ in this context). 

Done - Replaced discourse with "language and concepts"

10 Lines 158-159 why did you use two recruitment strategies – clinical and public – to simultaneously obtain a sample of eleven participants? What was the rationale? What was the added benefit of doing this? Please expand the rationale for this choice. 

Done - Added explanation to section on Inclusion Criteria and Recruitment "These two strategies were used to improve response rates and the heterogeneity of the sample."

Procedure: 

11 Lines 172-174 – who led the fieldwork and recruitment? 

Done - Recruitment lead is described in the Inclusion Criteria and Recruitment section (2nd paragraph) and fieldwork lead is described in the procedure section (1st paragraph). Initials of authors are now unblinded. 

12 To add transparency, please list the initials of the research team involved in discussing emerging themes and reviewing fieldwork also earlier on (in addition to line 247). 

Done Author names and institutions have been unblinded 

13 Line 181 –what was the rationale behind having each workshop of three hours in duration? 

Done Stated that this was deemed adequate to cover objectives of workshops and refereed to S1 & S2 appendices with outlines of workshops. 

14 Lines 187-188 – were examples of photos provided to participants? Did any participant ask for further explanation of the photo task? If so, what did you say? 

Done Thank you for this suggestion. This is now addressed at the end of the Inclusion and Recruitment section, and the Participant Information leaflet (which included two sample images) is added as an appendix. And further description in the Strengths and Limitations section (End of paragraph 1)

15 What ethical issues were discussed? 

Done Thank you. I have expanded on the ethical issues that were covered as part of Group Workshop 1 and the details are included in the appendix. 

16 Use of consent forms for photographic subjects: what did the consent form include? What ethical guidance did you refer to and have used in this photovoice study? Please reference any guidance accordingly. 

Done - Consent forms included in S5 Appendix

17 Was the camera left to participants for 2 weeks? 

N/A It is mentioned at the end of the Ethical Considerations section that the cameras were gifted to the participants. 

18 Was there any follow-up conducted with participants during this time? 

Done - The lead researcher contacted participants by phone during the 2 weeks to check on their progress and schedule the interview (Procedure section / Interviews paragraph)

19 Did any of the participants encounter any challenges in taking photos during the 2 weeks? If so, what was and how did you resolve it? 

Done - No notable challenges were presented. Some technical issues with using cameras were experienced (now explained in the Strength & limitations section)

20 Why did you the decide to divide participants into groups of two to three (workshop 2)? 

Done - They were randomly assigned into two groups to ensure effective small group discussion (Procedure, 1st paragraph)

21 Lines 215-217 “Group B were asked to review photos following the same procedure as Group A, but at the end of the workshop they were presented with Group A’s themes to incorporate into their selection”. How did this work in practice? Please expand. 

Done - Thank you. This is now expanded upon in Procedure, Group Workshop 2.

22 Lines 226 - it is great to see the photo-exhibition organised. How many people attended the event? And who were the attendees? (e.g. policy makers, community members…) How many participants attended and presented their photos at the event? 

Done Thank you. We have expanded on the description of attendees at the exhibitions. There were numerous events and attendees were not required to sign in (although they were invited to), therefore only estimated and incomplete attendance figures could be generated. 

23 Were all participants involved in the analysis? Fig. 1 and Appendix should be explained also in the text, to assist the reader in understanding what was done. 

Done - Thank you for the suggestion. This is now included in the Strengths and Limitations section. 

24 In particular, explain how participants were involved in coding the photos and identify main themes, and how you have linked this to your analysis. How did you incorporate participants’ coding and identification of main themes in the thematic analysis that you conducted? It would be valuable to understand how these two elements were linked. This will support what covered in Appendix S2 and S3. 

Done - Thank you. We believe with the suggested improvements (listed above) to the manuscript, we now sufficiently expand on this process. 

25 The appendices the authors provide are very informative and detailed and add strength to the overall paper. However, I suggest to expand this section (see above questions), as it is important to present how photovoice methods were applied in this study. 

Done - Further appendices added (PIL and consent form)

26 Line 256 – please change the verb into the past tense 

Done- Thank you for pointing out this typo.

 Results 

27 Figure 3 – can you provide a quote supporting the explanation of this photograph? This comment applies to all photographs that don’t have a supporting quote. Although the authors explain the reported meanings associated with each photo (and in some cases, a supporting quote is presented), for transparency, each photo should be presented with the participant’s explanation as well. 

Done Thank you for the suggestion. Additional captions and quotes have been added to Figs 3 and 4. The caption with Fig 4 is particularly long which is why we initially chose to summaries it, but we appreciate it may be better to provide a first-order analysis if space allows. Reviewing the remaining images and quotes, the team are satisfied with the presentation of the results and quotations as they stand. 

28 Also, next to each quote, I recommend you include the participant number, gender, and age. Having this info helps the reader to contextualise the findings. E.g. (PA, F, 37). Otherwise, the quotes feel a bit difficult to follow and put into context. 

n/a The authors do not wish to link quotations to particular individuals unless essential, as this may undermine anonymity. We are not convinced of the added value that this information provides. We will ask editor for guidance. 

29 The same applies for the photos presented. Please specify which participant took each photo. 

n/a As above.

30 Line 628 – what is HCPs? 

n/a The acronym HCP is explained in the text where first mentioned. 

Strengths and limitations: 

31 I suggest the authors to support many of their statements with appropriate references. 

Done Thank you. This is done throughout as suggested by Reviewer 2.

32 Lines 870-872 – Please add how photovoice proved to be an effective tool for actively engaging individuals in reflecting and communicating the lived experience of self-managing RA. What did you observe? What did you find? 

Done Reflections on evaluating the methodology has been expanded upon in the Strengths and Limitations section

33 Lines 872-873 – please add why you felt that this was the case (do you have any evidence of this? What did you observe?). It would be good to comment further and expand this statement: “participants reported to find the process of personal reflection, and the sharing of mutual concerns and experiences rewarding”. As suggested above, there are some papers that have examined and assessed the value of using a combination of photos, interviews, and group discussions to access people’s lived experiences/perceptions. Please refer to examples made in Lines 95-97. 

Done Thank you. This comment has been clarified and expanded upon. Additional citations have been added.

---

## [Decision Letter · Decision Letter 1]

22 Feb 2021

(In)Visible Illness: A Photovoice Study of the Lived Experience of Self-Managing Rheumatoid Arthritis

PONE-D-20-20019R1

Dear Dr. Donnelly,

We’re pleased to inform you that your manuscript has been judged scientifically suitable for publication and will be formally accepted for publication once it meets all outstanding technical requirements.

Kind regards,

Andrew Soundy

Academic Editor

PLOS ONE

Additional Editor Comments (optional):

Reviewers' comments:

Reviewer's Responses to Questions

**Comments to the Author**

1. If the authors have adequately addressed your comments raised in a previous round of review and you feel that this manuscript is now acceptable for publication, you may indicate that here to bypass the “Comments to the Author” section, enter your conflict of interest statement in the “Confidential to Editor” section, and submit your "Accept" recommendation.

Reviewer #1: All comments have been addressed

2. Is the manuscript technically sound, and do the data support the conclusions?

Reviewer #1: Yes

3. Has the statistical analysis been performed appropriately and rigorously? 

Reviewer #1: N/A

4. Have the authors made all data underlying the findings in their manuscript fully available?

Reviewer #1: Yes

5. Is the manuscript presented in an intelligible fashion and written in standard English?

Reviewer #1: Yes

6. Review Comments to the Author

Reviewer #1: (No Response)

7. PLOS authors have the option to publish the peer review history of their article (what does this mean?). If published, this will include your full peer review and any attached files.

Reviewer #1: No

---

## [Editor Report · Acceptance letter]

26 Feb 2021

PONE-D-20-20019R1 

(In)Visible Illness: A Photovoice Study of the Lived Experience of Self-Managing Rheumatoid Arthritis 

Dear Dr. Donnelly:

I'm pleased to inform you that your manuscript has been deemed suitable for publication in PLOS ONE. Congratulations! Your manuscript is now with our production department. 

Kind regards, 

on behalf of

Dr. Andrew Soundy 

Academic Editor

PLOS ONE